# Synergistic Membrane Disturbance Improves the Antibacterial Performance of Polymyxin B

**DOI:** 10.3390/polym14204316

**Published:** 2022-10-14

**Authors:** Wenwen Li, Che Zhang, Xuemei Lu, Shuqing Sun, Kai Yang, Bing Yuan

**Affiliations:** 1Center for Soft Condensed Matter Physics and Interdisciplinary Research, School of Physical Science and Technology, Soochow University, Suzhou 215006, China; 2Songshan Lake Materials Laboratory, Dongguan 523808, China; 3School of Optical and Electronic Information, Suzhou City University, Suzhou 215104, China

**Keywords:** membrane disturbance, membrane permeabilization, nanocomposite, antimicrobial agent, drug-resistant bacteria

## Abstract

Drug-resistant Gram-negative bacteria pose a serious threat to public health, and polymyxin B (PMB) is clinically used as a last-line therapy for the treatment of infections caused by these pathogens. However, the appearance of PMB resistance calls for an effort to develop new approaches to improve its antibacterial performance. In this work, a new type of nanocomposite, composed of PMB molecules being chemically decorated on the surface of graphene oxide (GO) nanosheets, was designed, which showed potent antibacterial ability through synergistically and physically disturbing the bacterial membrane. The as-fabricated PMB@GO nanocomposites demonstrated an enhanced bacterial-killing efficiency, with a minimum inhibitory concentration (MIC) value half of that of free PMB (with an MIC value as low as 0.5 μg mL^−1^ over *Escherichia coli*), and a bacterial viability less than one fourth of that of PMB (with a bacterial reduction of 60% after 3 h treatment, and 90% after 6 h incubation). Furthermore, the nanocomposite displayed moderate cytotoxicity or hemolysis effect, with cellular viabilities over 85% at concentrations up to 16 times the MIC value. Studies on antibacterial mechanism revealed that the synergy between PMB molecules and GO nanosheets greatly facilitated the vertical insertion of the nanocomposite into the lipid membrane, leading to membrane disturbance and permeabilization. Our results demonstrate a physical mechanism for improving the antibacterial performance of PMB and developing advanced antibacterial agents for better clinic uses.

## 1. Introduction

Public health is now faced with serious threats from many malignant bacteria, especially Gram-negative ‘superbugs’ such as the multidrug-resistant *Acinetobacter baumannii* (MDRAB), Vancomycin-resistant *enterococcus* (VRE), and Methicillin-resistant *Staphylococcus aureus* (MRSA) [1,2,3]. Due to the potent activity, polymyxin B (PMB) is selectively predominant towards all Gram-negative bacterial species (except the Proteus groups) and generally used as a last-line therapy of patients infected with multidrug-resistant pathogens [4,5,6]. However, the increasing use of PMB gives rise to the appearance of polymyxin-resistant strains, which are even found in public water facilities [7]. Therefore, there is an urgent need to develop new strategies to improve the antibacterial performance of PMB. 

As a typical type of polycationic antimicrobial lipopeptides, PMB is characteristically composed of an N-terminal fatty acyl chain, a linear tripeptide segment, and a heptapeptide ring (Figure 1a). The antibacterial activity of PMB is tightly associated with its membrane-disruption ability, and different membrane-action mechanisms such as the self-regulated ‘ladder’ model and carpet model (especially at higher concentrations) are suggested [4]. It is generally accepted that lipid A, the main constituent part of lipopolysaccharide (LPS) in the outer membrane (OM) of bacteria, is one of the action targets of PMB [8,9]. The adsorption and aggregation of PMBs on the OM surface disrupt the high surface charge density of the counter-crosslinked lipid A moieties due to DAB residues of the peptides, and consequently facilitate penetration of hydrophobic fatty acid tails into the membrane, leading to destabilization and permeabilization of the OM [5,8,10]. A similar disruption process of the inner membrane (IM) is also established, composed of the DAB-based adsorption followed by a rapid insertion of fatty acid tails into the bilayer. In addition to the integrity loss of the bacterial membrane, the disruption of osmotic balance of bacteria due to the contact and lipid exchange between the OM and IM is also suggested, as a result of the translocation of PMB across the OM and the consequent disturbance of the IM [11]. On the other hand, the resistance of PMB has recently been reported with the emergence of mobile colistin resistance (mcr) genes, which mediates the addition of a phosphoethanolamine to a phosphate at the N-acetylglucosamine headgroup of lipid A in the OM [12]. Such a modification reduces the membrane charge gradient and increases lipid packing, which hinders the membrane binding and impairs the bactericidal activity of lipopeptide antibiotics [13,14]. Generally, it is indicated that the enhancement of membrane-disrupting ability is a key to improve the antibacterial performances of PMB.

Nanoparticles (NPs) with engineered structure provide a promising route to aid the membrane interaction of many antibacterial agents [15,16,17,18,19,20]. Golden, silver, and polymetric NPs are surficial-decorated with antibiotics or antimicrobial peptides, which show strengthened membrane-disrupting and bacterial-killing ability [18,19,21,22,23,24]. This enhancement is generally ascribed to the increased local density of positive charges and antimicrobial molecules [17,18,19,22]. Among NPs, graphene oxide nanosheets (GO) demonstrate great potential in biomedical applications due to advantages including good water dispersion, low biological toxicity, easy modification, and mass production [15,16,17,25,26]. Specifically, the unique two-dimensional structure and amphiphilic surface endow GO to penetrate the cell membranes like a “nano-knife” and even extract lipids from the outer leaflet, inducing physical disturbance to the membrane structure [15,17,27,28]. Therefore, it is reasonably speculated that this ability of GO could give a possibility to trigger the membrane-action of PMB for improved antibacterial performance.

In this work, using a covalent conjugate method [23], a type of PMB@GO nanocomposite composed of PMB molecules decorated on the GO (~100 nm) surface was fabricated, which showed enhanced membrane-disturbance and antibacterial efficiency compared to the original components, due to the synergistic antibacterial mechanism between PMB molecules and GO nanosheets in the composites. Simulations showed that the PMB decoration adjusted the interaction between GO and membrane, which drove the penetration of nanosheets in a vertical manner and meanwhile facilitated the insertion of PMBs into the bilayer. Such synergy between PMB and GO strongly enhanced the membrane-disrupting ability of the nanocomplex, which was clearly demonstrated by scanning electron microscopy (SEM) images and giant unilamellar vesicle (GUV) dynamic release tests. Furthermore, the nanocomposites exhibited moderate cytotoxicity at bactericidal concentrations, due to their membrane-selective action mechanism. This work provides a new possibility for improving the clinical applications of PMB based on physical membrane-disturbance strategies.

## 2. Materials and Methods

### 2.1. Materials

1,2-Dioleoyl-sn-glycero-3-phosphocholine (DOPC), 1,2-dioleoyl-sn-glycero-3-phospho-(1’-rac-glycerol) (DOPG) and 1,2-dipalmitoyl-sn-glycero-3-phosphoethanolamine-N-(lissaminerhodamine B sulfonyl) (RhB-PE) were purchased from Avanti Polar Lipids (Alabaster, AL, USA) and used as received. Calcein, cholesterol (Chol), polymyxin B (PMB), and N-hydroxysuccinimide (NHS) were obtained from Sigma-Aldrich (Darmstadt, Germany). Tetraethylenepentamine (TEPA) and 1-(3-dimethylaminopropyl)-3-ethylcarbodiimide hydrochloride (EDC) were obtained from Beijing Bailingway Technology Co., Ltd. (Beijing, China). Carboxyl-modified graphene oxide (GO) quantum dot dispersion was purchased from Nanjing XFNANO Materials Tech Co., Ltd (Nanjing, China). and sonicated for 30 min each time before use. All experiments were carried out at room temperature at 22 °C.

### 2.2. Synthesis and Characterizations

The GO nanosheet dispersion was treated with an ultrasonic cell crusher (JY92-IIN with #2 amplitude-change pole, Scientz, Ningbo, China) at 650 W for 20 min (2 s ON, 2 s OFF), and purified with washing–centrifugation cycles (14,000 rpm × 10 min) three times to a final pH value of 6.5, which was similar to that of water used for washing. The solution was stored at 4 °C in the dark. Right before use, the dispersion was centrifuged at 8000 rpm for 10 min. No obvious precipitation was observed after centrifuging, presenting a good dispersion performance of the nanosheets. The physical mixture of PMB and GO (termed as PMB + GO) was obtained by directly mixing the GO dispersion (1 mg mL^−1^ in ultrapure water) with PMB solution (1 mg mL^−1^ in ultrapure water) at a volume ratio of 1:8, and using it as mother solution in the tests. Each time before use, the solution was vortexed for 20 s for good dispersion.

The PMB@GO nanocomposite was prepared following the traditional EDC/NHS crosslinking method. TEPA (3 mM, pH = 5), EDC, and NHS were added to the GO dispersion (1 mg mL^−1^; 0.1 mL) and shaken at low temperature for 6 h for carboxyl activation. After that, PMB solution (1 mg mL^−1^; 0.8 mL) was added and shaken for another 3 h. The obtained complexes were dialyzed (MW 1 kD) to remove the excess chemicals and concentrated for further characterization. The obtained solution was stable without apparent precipitation at 4 °C in the dark for more than three months. The solution was vortexed for 20 s each time before use.

The dynamic light scattering (DLS; DynaPro, Malvern, UK), Fourier transform infrared spectroscopy (FTIR; Nicolet 6700, Thermo Scientific, MA, USA), and UV–vis spectroscopy (UV-3600, SHIMADZU, Japan) methods were used to characterize the nanocomposites. Pure PMB and pure GO were used as controls. Atomic force microscopy (AFM; MFP-3D-SA, Asylum Research, Santa Barbara, CA, USA) was performed in tapping mode in an atmospheric environment. The samples for AFM imaging were prepared by drop-casting a diluted suspension onto a cleaned mica substrate and drying at room temperature. Powder X-ray diffraction (XRD) pattern of the GO film deposited on a silicon wafer was gathered on a Bruker D8-Advance diffractometer (Saarbrucken, Germany). Data from 1° to 13° were collected.

### 2.3. Membrane Activity Test

Giant unilamellar vesicles (GUVs) were formed by the conventional electroformation method [15,17]. The lipid composition was PC/PG = 4:1 or PC/PG/Chol = 3:1:1 and was premixed with 1 wt% Rh-PE for fluorescent labeling. The interior of GUVs was filled with calcein, which was used to detect membrane permeability. After centrifugation (8000 rpm × 20 min), the GUVs were dispersed in a homemade chamber. Then, a certain concentration of PMB or PMB@GO was injected slowly, and fluorescence leakage from the GUVs was observed under a confocal microscope (LSM 710, Zeiss, Oberkochen, Germany) after 30 min of incubation.

### 2.4. Antimicrobial Activity

*E. coli* CGMCC 1.12883 was grown to midlogarithmic phase and diluted to an optical density of approximately 0.1 at a 600 nm wavelength. A colony-forming unit (CFU) assay was performed following a well-standardized protocol [17,22]. *E. coli* was pretreated with PBS (as a control), pure PMB, or PMB@GO nanocomposite for 10 h before carrying out the CFU tests. Three replicates were performed for each condition, and the average value was obtained for analysis. After recording the CFU value in each condition, the percentage of CFU_agent_ by different agents was calculated following the equation:
$${\mathrm{CFU}}_{\mathrm{agent}}\;\left(\%\right)=\frac{{\mathrm{CFU}}_{\mathrm{agent}}}{{\mathrm{CFU}}_{\mathrm{control}}}$$


The minimal inhibitory concentration (MIC) of the agent was determined using the broth microdilution method [16,22]. A volume of 100 µL broth, containing 10^5^ CFU mL^−1^ of *E. coli*, was added to each well of the 96-well plates. GO, PMB, and PMB@GO were added to a final concentration of 0.25, 0.5, 1, 2, 4, 8, or 16 µg mL^−1^ by gradient. The MIC value was recorded as the lowest concentration that killed 90% of the bacteria, as observed with a microplate reader (Infinite F50, TECAN, Mannedorf, Switzerland). All MIC values were the average of more than three independent experiments.

The antibacterial activity of PMB, the physical mixture of PMB and GO (termed as PMB + GO), and PMB@GO were tested by the agar well diffusion method. Approximately 1 mL of bacterial dispersion (10^6^ CFU mL^−1^) was sucked into the petri dish by pipette gun, and 20 mL of sterilized agar medium was poured into the petri dish. Four separated holes (about 6 mm in diameter) were drilled in the agar medium after the medium was solidified. Measures of 50 μL of PBS, PMB, PMB + GO (8:1 by wt), and PMB@GO solutions, with a peptide concentration of 16 μg mL^−1^, were added to the four holes, respectively. The plate was incubated for 24 h at 37 °C. After incubation, the diameters (mm) of the inhibition zone were measured. Each sample was assayed in triplicate. 

The live and dead cell staining test was performed following the traditional method. Briefly, *E. coli* was incubated with PBS, PMB (at its MIC concentration of 1.0 μg mL^−1^), and PMB@GO (at its MIC concentration of 0.5 μg mL^−1^) in an incubator (37 °C, Shanghai, China) for 12 h. The bacteria were harvested by centrifugation at 8000 rpm for 5 min and washed two times with PBS. After staining with propidium iodide (PI) and DAPI for 30 min at room temperature, the bacteria were washed twice and resuspended in 1 mL of PBS. Thus, 50 µL of bacterial suspension from each sample was uniformly spread on cleaned glass. In situ observation was performed under an inverted confocal microscope system (LSM 710, Zeiss, Oberkochen, Germany) equipped with a 63× oiled objective lens. All images were captured under the same instrumental settings.

### 2.5. Morphological Characterization of Drug-Treated Bacteria

Bacteria were treated with GO, PMB, and PMB@GO (at 5× MIC concentrations) and fixed with 2.5% glutaraldehyde overnight at 4 °C. After washing with PBS buffer, the bacterial cells were dehydrated through sequential treatments of 20%, 40%, 60%, 80%, and 100% ethanol for 30 min [16,22], dropped onto a cleaned silicon wafer to lyophilize, and imaged using a scanning electron microscope (SEM; SU8010, HITACHI, Tokyo, Japan) after gold coating.

### 2.6. Cytotoxicity Assay

The traditional MTT assay was performed to evaluate the cytotoxicity of the agents [17,29]. Briefly, human gastric cancer (MGC-803) cells were cultured to the middle logarithmic growth period, centrifuged at 3000 rpm for 10 min, moved to a 96-well plate (~10^4^ per mL, 100 µL in each well), and incubated overnight until all the cells adhered to the wall. After that, the agent was added (free PMB or PMB@GO, from 0.25 to 8 µg mL^−1^) and DMEM was used as the control. The cells were co-incubated with agents for 24 or 48 h, after which they were treated with MTT (5 mg mL^−1^ × 20 µL) for another 4 h. After being centrifuged for 10 min (3000 rpm), the precipitates were redispersed in 100 µL DMSO and recorded at 490 nm by a microplate reader. 

### 2.7. Hemolysis Assay

The hemolysis assay was performed as reported previously [17]. The red blood cell suspension was treated with free PMB or PMB@GO (from 0.25 to 16 μg mL^−1^) for 4 h at 37 °C. After that, the mixture was centrifuged at 5000 rpm for 10 min, and the OD value of the supernatant at 550 nm was measured. Ultrapure water was used as a positive control, and PBS served as the negative control.

### 2.8. Simulation Models and Methods

The MD simulations were performed by using the GROMACS 4.6.5 software package (Berendsen lab, University of Gottingen, Germany) with the MARTINI force field [17]. Non-bonded interactions, including electrostatics and Lennard–Jones, were cut off at 1.2 nm. All simulations were carried out in the isothermal–isobaric (NPT) ensemble at the temperature of 298 K and pressure of 1.0 bar. The temperature and pressure were maintained with the Berendsen temperature coupling scheme with a time constant of 1 ps, a Parrinello–Rahman semi-isotropic barostat with a time constant of 12 ps, and compressibility of 3 × 10^−4^ bar^−1^ [30]. A time step of 20 fs was used and periodic boundary conditions were applied in all three directions. 

The coarse-grained (CG) model of PMB was obtained by using the Martinize script (Prof. dr. Siewert-Jan Marrink, University of Groningen, Groningen, Netherlands) with the crystal structure of peptide [10,31,32]. For GO nanosheets, the nanosheet was constructed by mapping 9 carbon atoms into 1 CG SC4 bead, and the beads within 1.2 nm were constrained by harmonic springs with a force constant of 12,000 kJ mol^−1^ nm^−2^. Moreover, some SC4 beads were further randomly chosen and replaced with a P1-type bead to represent the functional groups of GO, as performed in previous studies [33]. The lipid bilayer used in the simulations consisted of 512 DOPC lipid molecules. In addition, 0.1 M NaCl was included in the system and additional Cl^−^ ions were added to neutralize the charge on the peptides.

## 3. Results and Discussion

### 3.1. Characterization of PMB@GO

The GO nanosheets, with a size distribution of 30~50 nm and a mean thickness of 2.1 ± 0.5 nm (Figure 1e), were well-dispersed in aqueous solution without obvious aggregation for more than three months. Appendix A shows the powder X-ray diffraction (XRD) profile of a GO film deposited on a silicon wafer, which shows an inter-layer spacing between GO nanosheets of ~7.7 Å. This indicates a layer number of between two and three for each GO nanosheet. Using the traditional EDC/NHS crosslinking method, PMBs were chemically decorated on the GO surface through the carboxyl–amino coupling between them (Figure 1a). For the preservation of its antibacterial activity, an excessive dose of PMB was adopted (with GO: PMB = 1: 8 by weight) to retain as many active amino groups on it as possible [5,31,34,35], and the dosage of PMB in the composite system was used to record the concentration of the PMB@GO in this work. The PMB@GO nanocomposite was characterized using a variety of techniques to confirm the successful decoration of PMBs to GO (Figure 1b–e). The UV–vis absorption spectra (UV–vis) show a wide absorption band of the pure GO nanosheets and a characteristic bond at 300~400 nm of the pure PMB [36]. After decoration, a left-shift of the bond is observed (at 278 nm). Fourier transform infrared spectroscopy (FTIR) shows almost no difference among the nanocomposite, GO, and PMB, as the newly synthesized bond between carboxyl and amino groups overlaps with that of the original PMB (C=O at 1640~1700 cm^−1^ and N-H/C-H at 3150~3520 cm^−1^) and GO (C=O at 1640~1700 cm^−1^ and O-H/C-H at 3150~3520 cm^−1^). The most notable evidence comes from the size determination under dynamic light scattering (DLS) and atomic force microscopy (AFM) tests. DLS shows an increased hydration diameter of the nanocomposite (with a mean size of 453.6 nm) in comparison with the original GO of 129.0 nm, while AFM demonstrates a mean lateral size of 48.2 ± 21.5 nm with a thickness of 4.1 ± 0.6 nm of the PMB@GO nanocomposite in comparison with the much smaller GO nanosheets. Thus, it is suggested that the PMB@GO nanocomposite normally has monodispersed and layered structure with PMB molecules decorated on the sheet surface. In contrast, the physical mixture of PMB and GO (i.e., PMB + GO) shows similar DLS and AFM results as that of GO, due to the good dispersion of PMB as small molecules in the solution. 

### 3.2. Growth Inhibition and Membrane Destabilization of PMB@GO over Bacteria

To evaluate the bacterial inhibition effect of PMB@GO nanocomposite, the agar disk-diffusion test was first performed. A typical type of Gram-negative bacteria, *Escherichia coli,* was used as an example. The agar hole diffusion-inhibition zone method can conveniently detect the antibacterial effect of drugs. Holes (about 6 mm in diameter) filled with 50 μL of PBS, free PMB, the physical mixture of PMB and GO (at 8:1 by wt, termed as PMB + GO), and PMB@GO nanocomposite (peptide concentration of 16 μg mL^−1^ with respect to the peptide), respectively, were located in the solidified agar medium pre-inoculated with lawns of *E. coli* bacteria. Herein, no positive control was used, as PMB has been generally employed as a positive control in related research due to its potent antibacterial ability towards most Gram-negative species. After 24 h incubation, diameters of the inhibition growth zones were measured. As a control, PBS does not show any bacterial growth inhibition effect (Figure 2a). The inhibition zones of free PMB and PMB + GO are determined to be ~11.5 mm, while that of PMB@GO is ~12.5 mm, demonstrating an enhanced bacterial inhibition effect of the nanocomposite. 

For a quantitative evaluation of the bacterial inhibition ability, the minimal inhibitory concentration (MIC) test was performed (herein, the lowest concentration that kills 90% of the bacteria was recorded as the MIC value). As shown in Figure 2b,c, although the MIC value of free PMB is as low as 1.0 μg mL^−1^, that of the PMB@GO nanocomposite is even lower, being only half of it. In comparison, the GO used in this work does not show antibacterial effect at the corresponding concentrations. Furthermore, the physical mixture of PMB + GO does not show any improvement of the antibacterial ability of PMB either. To examine the effect of agents on the physiological function of bacteria (e.g., reproduction), the bacteria were treated with the agents at their 5× MIC concentrations for 3 h or 6 h, after which the bacteria were diluted and incubated 48 h (without the agent) before colony count. As shown in Figure 2d, the free PMB exhibits a 60% reduction in bacteria after 3 h treatment, whereas for PMB@GO, this percentage is over 90%. After 6 h incubation, almost all cells are inactivated by PMB@GO. This suggests an advanced bacteriostatic effect of the PMB@GO nanocomposite. 

Additionally, the live and dead cell staining test was carried out and imaged under a confocal laser scanning microscope (CLSM). The nucleic acid dye DAPI was used to stain both live and dead cells while propidium iodide (PI), a nucleic acid dye that can only penetrate damaged membranes, was used to mark dead cells. After 12 h of incubation at the MIC concentrations (i.e., 1.0 μg mL^−1^ for free PMB, and 0.5 μg mL^−1^ for PMB@GO; PBS was used as the control), cell membrane damage was detected on over 70% of the PMB@GO-treated bacteria (representatively shown in Figure 2e). In contrast, only 46% of the bacteria were inactivated after PMB treatment. SEM image (at 5× MIC concentrations) confirms the destabilization and even physical damage of PMB@GO nanocomposite to the pathogen membrane. Significant deformation and even peeling off of membranes are observed in the bacteria after treatment with PMB@GO (marked with white arrows, as representatively shown in Figure 2f), which causes obvious cytoplasmic leakage. Such disturbance is much stronger than that generated by free PMB or GO, indicating a synergistic enhancement effect between PMB and GO components in the composite. Generally, all these results clearly demonstrate an improvement in the antibacterial and membrane-destruction activity of the nanocomposite over the Gram-negative bacteria, compared with that of free PMB or GO.

### 3.3. Hemolysis and Mammalian Cytotoxicity of PMB@GO Nanocomposite

The hemolysis and mammalian cytotoxicity of PMB@GO nanocomposite is a critical parameter to evaluate its clinical application. Figure 3 demonstrates that the toxicity of PMB@GO nanocomposite is comparable to that of free PMB. Specifically, the standard MTT assays performed on Human gastric cancer (MGC)-803 cells show that PMB@GO treatment, up to a concentration of 8.0 μg mL^−1^ (i.e., 16 times the MIC concentration), decreases the cellular mortality by less than 15% after 24 h of incubation, being comparable to that of free PMB at the same concentration (i.e., eight times its MIC value). Moreover, a prolonged incubation of 48 h does not induce an obvious difference. Meanwhile, the hemolysis was evaluated by incubating red blood cell stock with the composite, as well as free PMB, at different concentrations. A dose-dependent relationship between concentration and cytotoxicity of the composite is observed, being similar to that of free PMB. However, insignificant hemolytic activity (<10%) is observed in the composite up to a concentration of 16.0 μg mL^−1^ (i.e., 32 times its MIC value; Figure 3c and Appendix A). This result suggests a bacterial-membrane-selective action mechanism of free PMB, which is maintained after GO decoration for the PMB@GO nanocomposite.

### 3.4. Analysis of the Physical Membrane-Disturbance Mechanism

To further understand the membrane activity of PMB@GO nanocomposite over different membranes (i.e., the bacterial and mammalian cell membranes), the giant unilamellar vesicle (GUV) leakage assay, which has been generally used to evaluate the membrane permeabilization effect of agents, was performed [15,16,17,37]. GUVs containing calcein, a water-soluble and membrane-impermeable fluorophore, were prepared following the traditional electro-formation method [15]. Specifically, the GUVs consisting of neutral 1,2-dioleoyl-sn-glycero-3-phosphocholine (DOPC) and negatively charged 1,2-dioleoyl-sn-glycero-3-phospho-(1’-rac-glycerol) (DOPG) in a 4:1 ratio were used to mimic the bacterial cell membranes as performed in previous studies [38,39]. In contrast, cholesterol (Chol), which is a key membrane component of mammalian cells and is absent in bacterial membranes, was involved to prepare the GUVs, mimicking the mammalian cell membranes (with PC:PG:Chol = 3:1:1 by mol). All the GUVs, labeled with 0.1 wt% 1,2-dioleoyl-sn-glycero-3-phosphoethanolamine-N-(lissamine rhodamine B sulfonyl) (RhB-PE), kept stable without calcein leakage for more than four hours under CLSM observation. However, after the addition of agents above threshold concentrations, obvious leakage of calcein was observed. Figure 4a,b show representative images of GUVs after agent incubation for 30 min. For the PC/PG membranes, the addition of PMB at a concentration of 1.0 μg mL^−1^ induces obvious calcein leakage to half of the GUVs in the system, indicating agent-induced membrane permeabilization. In contrast, for PMB@GO, the threshold concentration is reduced to 0.5 μg mL^−1^ (at which 60% of the vesicles leak). Briefly, these results suggest an enhanced membrane-permeabilization ability of PMB@GO compared with free PMB, which might be the reason of the enhanced antibacterial ability of the composite. 

Furthermore, for the PC/PG/Chol membrane, the threshold concentration of both PMB and PMB@GO is significantly enhanced, to four or five times that over a PC/PG membrane. Histograms of the threshold concentration values are shown in Figure 4c. Such a deviation of the agent activity between PC/PG and PC/PG/Chol membranes might be ascribed to the stable and rigid membrane structure due to the much-compacted packing state of lipids in a Chol-involved bilayer [40,41,42,43,44]. That is, the PMB@GO nanocomposites, as well as PMB, have a stronger permeabilization ability to the bacterial membranes over the mammalian cell membranes at the same concentrations, probably due to their physical membrane-disturbance mechanism.

Moreover, it is interesting to find that, although free GO induces moderate permeabilizing effect to the membrane (with a threshold concentration of over 2 μg mL^−1^; Appendix A), defects of the lipid packing state might have been produced, leading to formation of lipid–calcein composite (probably with GO) in the bilayer. This is demonstrated as the bright green circle (i.e., calcein) overlaid with lipids after GO actions. A similar phenomenon is observed for the PMB + GO mixture, even with the more compact PC/PG/Chol membrane (Appendix A). This strongly suggests a different membrane action mechanism of PMB@GO compared with free GO due to the PMB decoration on the nanosheets.

The antibacterial and GUV-permeabilization results above suggest a synergistic effect in membrane action between PMB and GO in the PMB@GO nanocomposite system. To give an insight into the underlying mechanism, MD simulations were performed to investigate the interaction process between agents and a bilayer membrane. Here, a well-established coarse-grained (CG) Martini force field was applied in the simulations [10,17]. It is found that the GO nanosheet tends to insert into the membrane quickly (within 2000 ns) and then be stably sandwiched between the two leaflets, whatever the initial orientations are (even when it is originally vertical to the membrane plane; Figure 5b and Appendix A). On the other hand, the free PMB tends to embed shallowly to the membrane (Figure 5c and Appendix A). These results are consistent with previous observations [10,45]. Moreover, after the embedding of GO or PMB, the membrane keeps intact without obvious disturbance of the bilayer structure. 

In contrast, the PMB@GO nanocomposite exhibits a more complex membrane action process. The nanosheet displays continuous swing and rotation above the membrane, with random contacts with the bilayer surface. This process is representatively demonstrated as the fluctuating values of cos (θ) between −1 and 1, where θ refers to the angle between the nanosheet and the membrane plane (Figure 5d). Meanwhile, the value of ΔZ, referring to the minimum distance between nanosheet and bilayer center, also shows strong fluctuations. However, after a critical point at ~4200 ns, the value of ΔZ drops dramatically, indicating an insertion of the nanosheet into the bilayer. The snapshots show that a direct contact between the edge of the nanosheet and the membrane surface is needed for the insertion (Figure 5a). It is interesting to find that, after this critical point, the value of cos (θ) keeps constant at ~0, suggesting an equilibrium state of the nanosheet, with an orientation being perpendicular to the bilayer plane (Figure 5d, orange background). This orientation might be ascribed to the PMBs decorated on the sheet surface, which are found locate at the headgroup–tail interface of lipids in the bilayer. This explains the different action mechanism of GO nanosheets without and with PMB decoration. Moreover, the swing and rotation of nanosheets above the membrane before insertion can be regarded as a “searching” process (Figure 5d, blue background), which in turn reflects the synergistic effect between PMB and GO nanosheets in the membrane action process. Specifically, the vertical insertion of the nanocomposite amplifies the defects on the membrane (Appendix A), which might facilitate the membrane permeabilization and be helpful for the following action of the PMB component.

GUV leakage assay and MD simulations suggest the synergistic effect between PMB and GO during the membrane actions of PMB@GO nanocomposites. MD simulations suggest that the GO nanosheet tends to swing and rotate above the membrane surface (regarded as the “searching” period) until a direct contact between the edge (or tip) of the nanosheet and the membrane. Like a knife, the GO inserts into the bilayer, which facilitates the embedding of PMB molecules (decorated on the GO surface) into the lipid headgroup–tail interface of the bilayer. The PMB molecules, in turn, justify the orientation of GO in the membrane, making it vertical to the bilayer plane. Such a physical cooperative insertion mechanism not only produces obvious structural defects of the membrane (demonstrated as the enhanced membrane permeability in the GUV assay), but also triggers the embedding of PMB into the membrane (especially against the PMB-resistant bacteria which have been mutated in lipid packing [13,14]) for the following biological actions. This working mechanism is different from that of pristine PMB or GO, which might be the reason for the enhanced membrane permeabilization and antibacterial effect of the PMB@GO nanocomposite. This mechanism is also different from that of abundant previous work on antibiotic–nanoparticle composites [46,47], in which the particles mainly play a role in increasing the local number density of antibiotic molecules.

## 4. Conclusions

In summary, an efficient approach was proposed to improve the antibacterial performance of PMB, an ‘ancient’ and clinical polypeptide drug. Through chemically binding the PMB molecules on the surface of GO, a typical and widely used 2D nanomaterial, the as-obtained PMB@GO nanocomposite exhibits an enhanced sterilization and bacteriostasis capacity against *E. coli*, with an MIC value half that of free PMB, and a bacterial viability less than one fourth that of PMB after an incubation of 3 h or 6 h. LIVE/DEAD cell staining and SEM observations show that the improved antibacterial ability is ascribed to the enhanced physical membrane-disrupting ability of PMB@GO compared with pristine PMB or the physical mixture of PMB and GO at the same peptide concentration. Moreover, PMB@GO shows moderate cytotoxicity or hemolysis effect at concentrations up to 16 times the MIC value (similar to that of clinical PMB), which makes the nanocomposite a potential candidate as an advanced antibacterial agent. A synergistic antibacterial mechanism between PMB and GO is revealed through combining the GUV leakage assay and MD simulations. This work provides an alternative candidate of PMB through decorating it on GO nanosheets for improved antibacterial performance, and reveals the underlying physical membrane-disturbance mechanism.

## Figures and Tables

**Figure 1 polymers-14-04316-f001:**
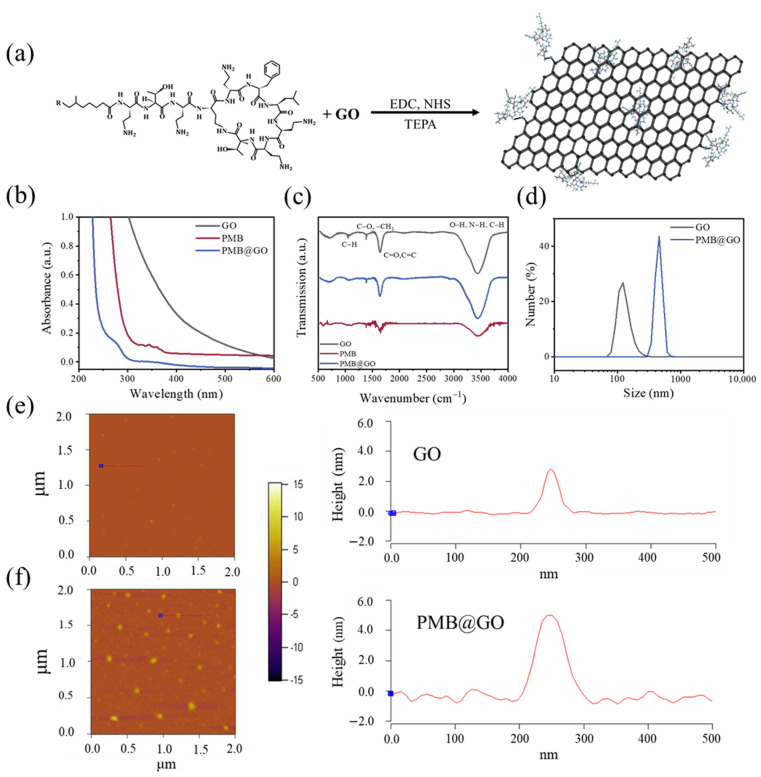
Synthesis and characterization of PMB@GO nanocomposite. (**a**) Chemical structure formula of PMB and schematic diagram of the PMB@GO nanocomposite synthesized by the EDC/NHS crosslinking method. Tetraethylenepentamine (TEPA) was used as catalyzer. (**b**–**e**) UV–vis absorption (**b**), FTIR (**c**), DLS (**d**), and AFM (**e**–**f**) characterizations of the nanocomposite. The results of pristine PMB and GO are also shown for reference. Height profile along the red line in AFM image is shown in the corresponding right panel.

**Figure 2 polymers-14-04316-f002:**
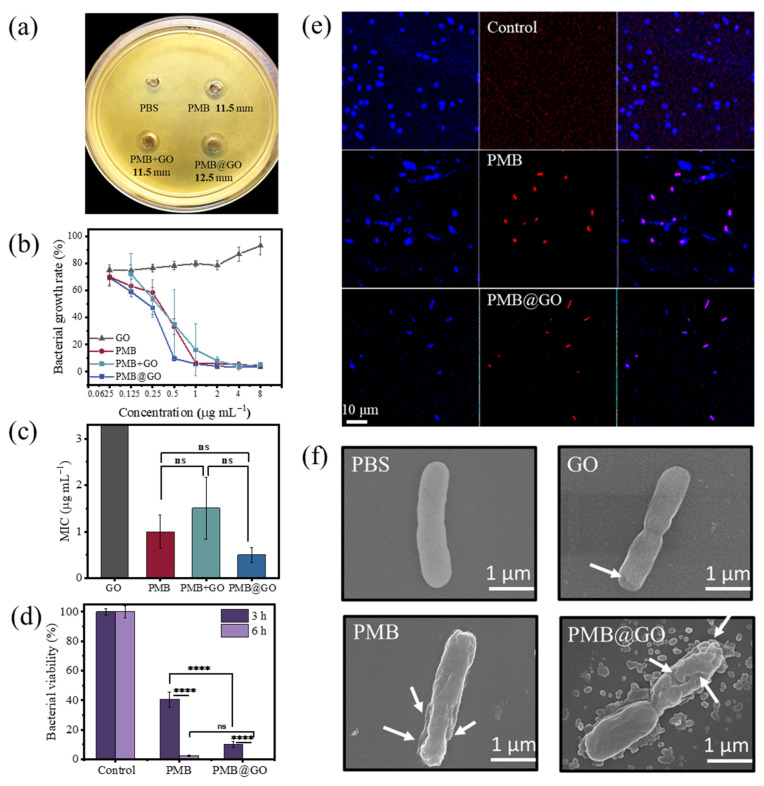
Enhanced antibacterial activity of the PMB@GO nanocomposites. (**a**) Inhibition zones of PMB@GO against *E. coli* in the agar hole diffusion-inhibition zone method. The assay performed with phosphate buffer solution (PBS) is used as the control. The conditions of free PMB and physical mixture of PMB and GO (i.e., PMB + GO) are also shown for comparison (at 16 μg mL^−1^ in terms of peptide). (**b**,**c**) Concentration-dependent bacterial viability (**b**) and MIC determination (**c**) of PMB@GO against *E. coli*. Free PMB, GO, and PMB + GO are shown as the control. *p*-values were calculated by t-test and ANOVA (ns means *p* > 0.05). (**d**) Proliferation capacity of *E. coli* after treatment with PMB and PMB@GO for 3 h or 6 h, respectively, with PBS as the control. *p*-values were calculated by t-test and ANOVA (**** *p* < 0.0001). (**e**,**f**) LIVE/DEAD staining (**e**) and SEM (**f**) images of microorganisms after treatment with PMB or PMB@GO at their 5 × MIC concentrations for 3 h. In (**e**), all bacteria: blue, dead bacteria: red, scale bar: 10 μm. White arrows in (**f**) representatively show the areas of heavy damage. PBS and GO (at 5 μg mL^−1^) are used as the control. *E. coli* was used as the test microorganism in these experiments.

**Figure 3 polymers-14-04316-f003:**
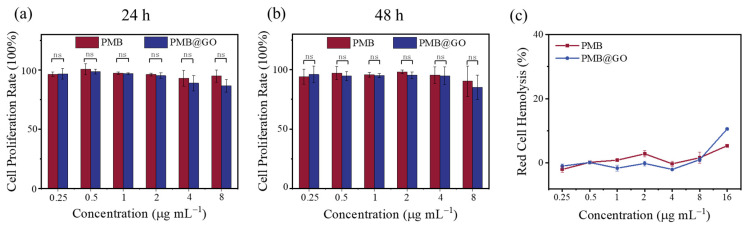
Toxicity tests of free PMB and PMB@GO nanocomposites. (**a**,**b**) Standard MTT cell viability assay of mouse gastric cancer (MGC)-803 cells cultured with free PMB or PMB@GO for 24 or 48 h. The concentrations of the agents varied between 0.25 and 8.0 μg mL^−1^. *p*-values were calculated by ANOVA (ns means *p* > 0.05). (**c**) Hemolytic activity of the agents. Red blood cells in PBS and ultrapure water were used as negative and positive controls, respectively, for determination of the hemolysis percentage values. The data show the normalized mean and standard error of at least three independent experiments.

**Figure 4 polymers-14-04316-f004:**
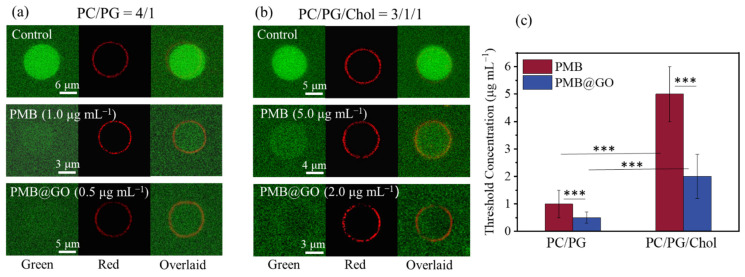
Membrane permeabilization test of free PMB and PMB@GO over different model membranes. (**a**) Representative confocal images of calcein-encapsulated GUVs (composed of PC and PG, with PC:PG = 4:1 by mol) after 30 min of incubation with no drug, PMB (at 1.0 μg mL^−1^), or PMB@GO (at 0.5 μg mL^−1^ with respect to peptide). (**b**) Representative confocal images of calcein-encapsulated GUVs (composed of PC, PG, and Chol, with PC:PG:Chol = 3:1:1 by mol) after 30 min incubation with no drug, PMB (5.0 μg mL^−1^), or PMB@GO (2.0 μg mL^−1^). (**c**) Histograms showing the threshold agent concentration for calcein leakage. Images in (**a**,**b**) were taken in the green (calcein), red (RhB-PE), and merged channels. The data in (**c**) show the normalized mean and standard error of five independent experiments. *p*-values were calculated by ANOVA (*** *p* < 0.001).

**Figure 5 polymers-14-04316-f005:**
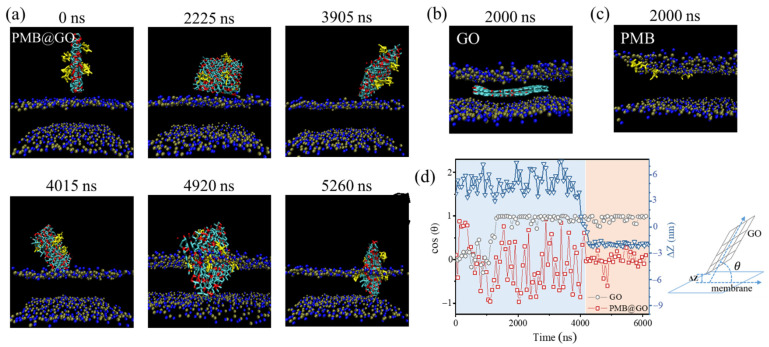
MD simulations of the membrane interaction with PMB@GO, free GO, or PMB. (**a**) Snapshots showing the insertion dynamics of PMB@GO into a membrane. (**b**,**c**) Snapshots showing the membrane interaction state of GO or PMB at the end of simulations (2000 ns). (**d**) Corresponding time evolution of ΔZ and cos (θ) of the PMB@GO and free GO during the membrane actions. Definition of ΔZ and θ is shown on the right. Blue and grey beads in snapshots: phospholipid headgroups; yellow rod: PMB; light blue sheet with red beads: GO with hydrophilic groups. A membrane consisting of 512 lipids is performed and lipid tails are not shown for clarity. The GO sheet in (**a**) has a size of 7.2 × 6.3 nm and is decorated with 3 PMBs. The GO in (**b**) has a similar size as that in (**a**). In (**c**), three PMBs are added.

## Data Availability

The data presented in this study are available in this same article.

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
