# Peer review of "Synergistic Membrane Disturbance Improves the Antibacterial Performance of Polymyxin B"

_polymers, 2022, doi:10.3390/polym14204316_

Round 1

Reviewer 1 Report

The authors reported the manuscript entitled "Synergistic membrane-disturbance improves the antibacterial performance of polymyxin B" very well. The authors discussed a type of PMB@GO nanocomposites composed of PMB molecules decorated on the GO (~100 nm) surface fabricated,  which showed enhanced membrane disturbance and antibacterial efficiency compared to the original components.  The manuscript has well written and well organised but the authors must consider some points to accept for publication;

1. English language of the manuscript must be checked.

2. Novelty and objective of the study should be clearly mentioned in the manuscript.

3. Please mention the model and company name of the instruments.

4. Discuss the role of GO. Compare the data with the published articles.

5. Conclusion section should be concise.

Author Response

Please refer to the separate file.

Reviewer 2 Report

The paper entitled „Synergistic membrane-disturbance improves the antibacterial performance of polymyxin B” focuses on the characterization of a covalent conjugate - PMB@GO nanocomposite, its antibacterial efficiency, membrane-disrupting ability, and in vitro cytotoxicity. The materials and processes were examined by various and complementary examination techniques such as dynamic light scattering, Fourier transforms infrared spectroscopy and UV–vis spectroscopy, Atomic force microscopy (AFM), CFU tests and agar well diffusion method for antimicrobial activity, scanning electron microscopy (SEM), MTT assay for cytotoxicity and MD simulations.

The introduction refers to the aim of the study, the experimental part is consistently revealed and explained while the results are understandably submitted and sufficiently illustrated. The conclusion summarizes the aforementioned results. In my opinion, the paper should be interesting from a scientific and practical point of view.

I would like to recommend the publication of the paper publication after some changes concerning the following issues:  

1.               More quantitative values could be given in the abstract;

2.               To increase the reproducibility of the results, the equipment used for the characterization of GO nanoparticles and polymyxin B-GO conjugates has to be specified in section 2.2;

3.               All graphs in the manuscript should be made larger for better visualization;

4.               Since the dose ratio of GO: PMB = 1:8 by weight is adopted, what is the content of GO and PMB in the conjugate?

5.               More comments concerning the results shown in Figure 1 are needed.

6.               Giving more explanations about the synergistic effect between PMB and GO nanosheet in the membrane action process taking into account the MD simulations of the three (PMB@GO, free GO, and PMB) systems will be a welcome step.

7.                Where applicable, the obtained results can be compared with other similar studies;

8.               The authors should also present the short- and long-term stability and agglomeration of the PMB@GO nanocomposite in physiological conditions since these phenomena can influence the biological performance and activity of the conjugate in vivo.

Reviewer 3 Report

Dear Authors,

The manuscript ID: polymers-1938456-v1 entitled Synergistic membrane-disturbance improves the antibacterial performance of polymyxin B” written by Wenwen Li, Che Zhang, Xuemei Lu, Shuqing Sun, Kai Yang and Bing Yuan is very interesting.

Antimicrobial resistance represents an enormous global health crisis and one of the most serious threats humans face today. Some bacterial strains have acquired resistance to nearly all antibiotics. The majority of the WHO list is Gram-negative bacterial pathogens. Due to their distinctive structure, Gram-negative bacteria are more resistant than Gram-positive bacteria, and cause significant morbidity and mortality worldwide. I agree with the Authors that polymyxin B (PMB) is clinically used as a last-line therapy in the treatment of infections caused by these pathogens. However, PMB-resistant bacteria are more often isolated. Therefore, it is necessary to search for new compounds or composites with antimicrobial activity towards these bacteria.

The whole manuscript (Introduction, Materials and Methods, Results and discussion and Conclusions) is properly organized. Introduction contains general data on antimicrobial resistance, activity of polymyxin B, and nanoparticles. The purpose of the work is concrete. Appropriate materials and methods were used to perform these studies. Statistical analyzes were also performed. The obtained results are documented, summarized in the form of schemes, tables or figures and properly interpreted. Based on the results, discussion and conclusions were drawn. It is a well written article.

However, I have some suggestions in order to improve paper, which are the following:

1)   I will add that there is no information about the concentrations at which these compounds killed bacteria (MBC – Minimal Bactericidal Concentration) and bactericidal (MBC/MIC ≤ 4) or bacteriostatic (MBC/MIC > 4) effect. Publication would be more precisious with these additional informations. Please remember this in the next article.

2) Lines 156 and 163: „2.5. Morphological characterization of drug-treated bacteria” and „2.6. Morphological characterization of drug-treated bacteria” – the titles are the same, please correct them. The hemolysis assay – please present as a separate subsection;

3)   Other:

Lines 122, 124, 132, 147, 232, 276, 238, etc.: E. coli – italics;

Line 152: In situ – italics;

Line 160: 30 min[15, 20], Line 165: agents[16, 26]. etc. – words are sometimes combined, please make spaces;

Line 196: 3. Results and discussion – 3. Results and Discussion

In my opinion, the obtained results are very valuable, original and manuscript is worth publishing in „Polymers”.

With highest regards,

Reviewer 4 Report

The manuscript "Synergistic membrane-disturbance improves the antibacterial performance of polymyxin B" presents interesting data. Before publication, the authors need to answer some questions that arise after the reading of the work.

1. GO samples characterisation would benefit from other methods, such as XRD. I suggest, if possible, performing the XRD characterisation.

2. The author speak about some figures and tables noted as Sx, most probably supplementary material. The reviewer is unable to see the supplementary material. Please correct the text or provide the SM

3. Figure 3a,b - please provide a statistical interpretation of the results.

4. The antimicrobial assays would be more relevant if a positive control would be used (a classic antimicrobial agent)

 5. The obtained results should be further discussed by comparing with relevant literature data  

Round 2

Reviewer 2 Report

The authors have carefully addressed the reviewer's recommendations. The paper may be published in its present form.